# Transcranial Pulse Stimulation in Alzheimer’s: Long-Term Feasibility and a Multifocal Treatment Approach

**DOI:** 10.3390/brainsci15080830

**Published:** 2025-08-01

**Authors:** Celine Cont-Richter, Nathalie Stute, Anastasia Galli, Christina Schulte, Lars Wojtecki

**Affiliations:** 1Department for Neurology and Neurorehabilitation, Hospital zum Heiligen Geist, Academic Teaching Hospital of the Heinrich-Heine-University Duesseldorf, 47906 Kempen, Germany; celine.cont@hhu.de (C.C.-R.); anastasia.galli@artemed.de (A.G.); christina.schulte@artemed.de (C.S.); 2Institute of Clinical Neuroscience and Medical Psychology, Medical Faculty, Heinrich-Heine-University Düsseldorf, 40225 Düsseldorf, Germany

**Keywords:** Alzheimer’s disease, long-term data, neuromodulation, real-world data, shockwaves, transcranial pulse stimulation

## Abstract

Background/Objectives: Neuromodulation is under investigation as a possibly effective add-on therapy in Alzheimer’s disease (AD). While transcranial pulse stimulation (TPS) has shown positive short-term effects, long-term effects have not yet been fully explored. This study aims to evaluate the long-term feasibility, safety, and potential cognitive benefits of TPS over one year in patients with Alzheimer’s disease, focusing on domains such as memory, speech, orientation, visuo-construction, and depressive symptoms. Methods: We analyzed preliminary data from the first ten out of thirty-five patients enrolled in a prospective TPS study who completed one year of follow-up and were included in a dedicated long-term database. The protocol consisted of six initial TPS sessions over two weeks, followed by monthly booster sessions delivering 6000 pulses each for twelve months. Patients underwent regular neuropsychological assessments using the Alzheimer Disease Assessment Scale (ADAS), Mini-Mental Status Examination (MMSE), Montreal Cognitive Assessment (MoCA), and Beck Depression Inventory (BDI-II). All adverse events (AEs) were documented and monitored throughout the study. Results: Adverse events occurred in less than 1% of stimulation sessions and mainly included mild focal pain or transient unpleasant sensations, as well as some systemic behavioral or vigilance changes, particularly in patients with underlying medical conditions, with some potentially related to the device’s stimulation as adverse device reactions (ADRs). Cognitive test results showed significant improvement after the initial stimulation cycle (ADAS total improved significantly after the first stimulation cycle (M_pre = 28.44, M_post = 18.56; *p* = 0.001, d = 0.80, 95% CI (0.36, 1.25)), with stable scores across all domains over one year. Improvements were most notable in memory, speech, and mood. Conclusions: TPS appears to be a generally safe and feasible add-on treatment for AD, although careful patient selection and monitoring are advised. While a considerable number of participants were lost to follow-up for various reasons, adverse events and lack of treatment effect were unlikely primary causes. A multifocal stimulation approach (F-TOP2) is proposed to enhance effects across more cognitive domains.

## 1. Introduction

Alzheimer’s Disease (AD) stands as one of the most pressing health challenges of our time, characterized by its progressive cognitive decline and the absence of a definitive cure. As the “baby boomer” generation ages, the number of those affected by AD is projected to double every 5 years and is predicted to reach 152 million by 2050 [1].

Among the pursuit of innovative therapeutic interventions, transcranial pulse stimulation (TPS) emerges as a promising avenue, offering a non-invasive approach that uses short, repetitive shockwaves. While initial studies have shown encouraging short-term effects of TPS on cognitive functions in Alzheimer’s patients [2,3], the need for long-term investigations to comprehensively assess its safety and effects remains utmost. These studies on short-term effects revealed that only a minority of stimulation sessions were associated with side effects of mostly minor severity, with headaches being the most prevalent, thus claiming TPS as a safe method [2,3].

A recent randomized, sham-controlled clinical trial published by Matt et al. involving 60 participants, showed that TPS demonstrated significant secondary endpoint improvements in executive function and depression in younger patients under the age of 70 years, although the primary cognitive endpoint was narrowly missed [4]. This highlights the need for further studies to better assess the potential of TPS in Alzheimer’s treatment.

This paper embarks on a longitudinal analysis, delving into extended-term data to elucidate the enduring impact of TPS on cognitive performance in Alzheimer’s.

While the complete effect of TPS remains unclear, Popescu et al. found a reduction of cortical atrophy after TPS stimulation in Alzheimer’s patients [5]. This result was predominantly found in patients who also improved in neuropsychological testing. Given that a highly focal method is used in several brain areas, it is important to examine specific changes in cognitive domains, rather than solely relying on general neuropsychological scores and improvement, as observed in most conducted studies. The area specificity of TPS was shown by an fMRI study that examined visuo-constructive functions that correlate with areas that were not stimulated and showed a decrease after treatment, possibly due to disease progression without symptomatic treatment effect [6]. TPS has also been shown to improve depressive symptoms [3,7], which are common in AD. Generally, a disease progression meta-analysis for Alzheimer’s Disease showed that a worsening by 5 points in a neuropsychological test battery, called the Alzheimer Disease Assessment Scale (ADAS), can be expected over a year [8]. Considering this, an improvement, or a stable score for over a year, would be considered a benefit and a suggestion to use TPS as an add-on therapy.

The commonly used treatment protocol for TPS targets the precuneus, bilateral parietal lobe, and bilateral frontal lobe, but the protocol used in this paper involves the bilateral temporal lobe as well [3]. The importance of targeting the precuneus, a key node for the default mode network affected by Alzheimer’s Disease, was shown by a recent randomized, double-blind, sham-controlled study that applied repetitive transcranial magnetic stimulation over the precuneus and showed a delay of cognitive and functional decline [9]. Moreover, FDG-PET imaging studies demonstrate early regional hypometabolism of posterior brain areas, such as the parietal lobes, in patients with Alzheimer’s Disease [10]. Furthermore, prefrontal cortex regions play pivotal roles in cognitive functions associated with social, emotional, and motivational behaviors [11], making them crucial targets for treating AD. Research on functional architecture of memory, especially episodic memory, has identified the hippocampus and the anterior-temporal and posterior-medial system [12]. Therefore, our protocol targets the bilateral temporal lobe. Even though it has not been visualized on the MR-navigations system that TPS can reach as deep as the location of the hippocampus, from the mechanisms of action of TPS, it is assumed that temporal stimulation reaches that area. These targeted regions—especially the frontal and temporal lobes—are critically involved in memory and speech processing. Given the high pulse density in these areas, we expect more pronounced and lasting effects in these cognitive domains, as reflected in Hypothesis iv. While our study primarily focuses on cognitive outcomes, recent work has also discussed the potential of non-invasive brain stimulation to modulate affective or social-cognitive processes depending on protocol and stimulation targets. However, in the context of advanced neurodegeneration, our present aim was to address cognitive domains most consistently impaired in AD, such as memory, language, and orientation.

Cognitive decline in episodic memory develops faster in patients with dementia, compared to normal aging, where the decline in speed of information processing and executive functions develops first [13]. In normal disease progression, different symptoms can be seen at different stages of AD. In mild AD, symptoms like loss of concentration, orientation, and change in mood appear. Moderate AD includes increased memory loss, loss of impulse control, as well as trouble in reading, writing, and speaking [1]. Since memory dysfunction is not always the predominant cognitive symptom for patients with Alzheimer’s disease [14], it is important to look at different cognitive domains and how these are affected by a new add-on treatment method. Especially, language and visuospatial presentations are common symptoms in early Alzheimer’s progression [14] and should, therefore, be targeted. Unlike many previous TPS studies that have focused primarily on memory and visuospatial abilities, our study also includes also a structured analysis of other cognitive domains such as language and orientiation over time.

This article investigates the first long-term neuropsychological outcomes for over a year using the consistent treatment of TPS for patients with Alzheimer’s.

Hypotheses for the study were:

**Hypothesis** **1** **(H1).**
*TPS is safe with respect to the number and severity of adverse (device) events, A(D)E.*


**Hypothesis** **2** **(H2).**
*Cognitive scores improve after the first stimulation cycle.*


**Hypothesis** **3** **(H3).**
*Cognitive and affective scores stay stable after one year of treatment.*


**Hypothesis** **4** **(H4).**
*A stronger effect can be seen in memory and speech compared to visuo-construction due to the targeted neuroanatomical treatment areas.*


## 2. Materials and Methods

### 2.1. Participants

Initially, 35 patients completed the first stimulation cycle, which consisted of six sessions within two weeks. See our previous publication for the short-term effects [3]. During the follow-up period, there was a gradual loss of participants due to various reasons, including discontinuation or unavailability (see Figure 1).

Importantly, some patients are still actively undergoing treatment and are currently in the 4–6 month or 7–9 month follow-up periods. This reflects that the data is dynamic, and the number of patients in follow-up stages might continue to evolve as treatments progress.

For this paper, ten patients (eight male, two female, age range 59–79 years, M = 70.7) diagnosed with Alzheimer’s Syndrome meeting the criteria outlined by clinical evaluations, MRI and CSF (NIA-AA criteria) treated over a year with TPS at the Department of Neurology and Neurorehabilitation at Hospital zum Heiligen Geist in Kempen, Germany, were included in the long-term database. Demographics of those patients who were included in the analysis are illustrated in Table 1. Participants were selected based on a defined Alzheimer’s Clinical Syndrome, grounded by a progressive change in memory function (using scores from the Mini-Mental Status Examination) and impairment of daily living for > six months. Exclusion criteria were relevant intracerebral pathologies unrelated to Alzheimer’s disease, such as vascular encephalopathy Fazekas grade 3, thrombosis in the treatment area, tumors, vascular malformations, cerebral amyloid angiopathy (CAA), and a history of or ongoing antibody therapy. Additionally, patients with blood clotting disorders or on oral anticoagulation, those who had received corticosteroid treatments within six weeks prior to the first application, or those with a history of epilepsy (defined as multiple seizures or a seizure with a focus) were excluded. Patients with pacemakers not approved for TPS therapy, recurrent syncope, severe everyday-affecting behavioral disturbances such as aggression or psychosis, or those who were pregnant were also excluded. Lastly, any medical condition that would prevent adherence to the protocol was an exclusion criterion.

The prospective inclusion in a registry of all TPS-treated patients was part of the local registry approved by the Ethics Committee of the Regional Medical Chamber (Ärztekammer Nordrhein, Nr. 2021026).

### 2.2. Transcranial Pulse Stimulation (TPS)

For the stimulation, the Neurolith © TPS device from Storz Medical was used, which allows neuronavigation using individual 3D T1 isometric voxel MRI scans. The treatment protocol involved the application of TPS using specific parameters: 4 Hz, 0.20 mJ/mm^2^, and targeting the bilateral frontal cortex, bilateral lateral parietal cortex, extended precuneus, and the bilateral temporal cortex. The TPS sessions were conducted over a period of 40 min, starting with six initial sessions within two weeks every second day and then administered once a month with one session (see procedure in Figure 2). For each session, 6000 pulses were applied. The procedure was carried out in accordance with the initial protocol and by trained professionals.

### 2.3. Adverse Device Events (ADE)

AEs were reported using a numeric rating scale (NRS), asking for the severity on a 0–10 scale (with 10 being the highest intensity). The patient and/or caregiver were asked before every treatment if any unintended medical occurrence had happened after the last session and advised to inform the clinic if anything occurred within the next hours or days after the treatment. Some patients were admitted as inpatients at the hospital for part of the stimulation, during which they were consistently monitored for side effects and closely observed. Adverse device effects (ADEs) were considered a specific subset of AEs related to the use of the investigational device. ADEs include any adverse events resulting from device malfunctions, misuse, or inadequate instructions for use, setup, implantation, installation, or functioning. In line with this definition, ADEs were regarded as adverse device reactions when a causal effect on the patient’s body was assumed or determined. ADEs can also occur in cases of device malfunction or misuse without direct biological effects.

The principal investigator (LW) assessed the causality of both AEs and ADEs based on medical data.

### 2.4. Neuropsychological Assessment

Participants underwent comprehensive neuropsychological evaluations pre- and post-TPS treatment, as well as every 3, 6, and 12 months (each ± 1 month). Standardized tests were used, such as Mini-Mental Status Examination (MMSE), Montreal Cognitive Assessment Test (MoCA), Alzheimer Disease Assessment Scale (ADAS), and its cognitive score (ADAS Cog). Moreover, the Beck Depression Inventory (BDI-II FS) was used to evaluate affective scores in some patients, which is a 21-question multiple-choice self-report scale that measures the severity of depression. The ADAS was used, focusing on multiple cognitive domains:Memory: Patients were presented with flashcards containing 10 words, which they were instructed to read aloud and memorize. Subsequently, they were prompted to recall the words they could remember (free reproduction). Following this, both new words and words from the initial list were presented, and patients were tasked with identifying the words they had previously learned (word recognition).Visuo-construction: The patients were provided with four distinct figures (circle, route, rectangles, and cube), each increasing in complexity, and were instructed to draw them.Orientation: Patients were asked regarding temporal and local orientation, including their first and last names, the date, the name of the hospital, and the city.Language: Patients were assessed regarding their speech expression, speech comprehension, and word-finding disorder during the entire assessment. This score was the sole measure evaluated subjectively (observer biased).

All assessments were conducted by trained study staff. Raters were not blinded to the study hypothesis.

### 2.5. Data Analysis

Quantitative analysis was conducted to compare pre- and post-TPS treatment neuropsychological scores using one-sided *t*-tests calculated with alpha = 0.05 for significance to assess improvement after the first stimulation cycle. Changes in cognitive performance within each domain were assessed, and correlations and regressions between time and cognitive outcomes were explored to test a stable score over one year. Significance was established at *p* < 0.05.

## 3. Results

### 3.1. Hypothesis 1 (H1): TPS Is Safe with Respect to Number and Severity of Adverse (Device) Events, A(D)E

In 187 sessions, 9 AEs were reported with a subjective severity rating by the patient ranging from 1.5 to 10 and a mean severity of *M* = 5.5 during the entire year of treatment (average treatment sessions were *M* = 18.7 over a year). Most of the AE were seen within the first stimulation cycle, where six sessions were conducted within two weeks, every second day (Table 2).

AEs were either focal or systemic. The latter were either vigilance changes or behavioral fluctuations such as nervousness or aggressiveness. Of the nine reported adverse events, seven of the AEs were considered as clear ADE related to stimulation, thus adverse device reactions. Five ADEs occurred during stimulation, including nervousness, feeling of pressure/slight pain on the head, pain in the jaw, and an unpleasant sensation on the head, syncope, and vomiting with hypotension. None of these ADEs/reactions lasted longer than the stimulation period. The remaining two AEs were judged to be unrelated to TPS and instead attributable to pre-existing medical conditions (e.g., cardiovascular issues).

Four AEs occurred after stimulation (somnolence with hypotension after stimulation, aggressive behavior and vigilance fluctuations for three days, and dizziness for two hours after treatment) and were partly related to stimulation (ADE, reactions) and partly to underlying medical conditions.

Two AEs were severe and were at least possibly partly related to stimulation (ADE, reactions) and partly to underlying medical conditions. Careful medical evaluations, including EEG, ECG, echocardiography, CT, blood sugar, and blood pressure measurements, identified hypertension as a related medical condition in severe A(D)E.

The first case was a patient (ID 3) who experienced a brief syncopal event during stimulation with accompanying hypotension and nausea, whereupon stimulation was terminated. Blood pressure was stable again a few hours later. This event could be considered an adverse device reaction (ADR), as it occurred in close association with the device and its stimulation.

The second case (ID 4) was a patient with a history of hypertension, COPD, and right heart failure who already felt unwell before the stimulation. After medical consultation, stimulation was performed, after which the patient appeared somnolent. Blood pressure was 100/60. The next day, the blood pressure was stable again. This case might also be regarded as an adverse device reaction (ADR), as the somnolence and hypotension could have been related to the stimulation, despite the patient’s underlying medical conditions.

Given that these two patients were already hospitalized, they could be consistently monitored as part of their ongoing care.

Summing up, hypothesis 1 (H1) can be verified with respect to the number of AEs below 1% of treatment sessions. However, AEs that at least have to be partly attributed to stimulation as ADE can not only occur focally as pain or unpleasant sensation but also systemically as behavioral changes or vigilance changes, especially in patients with underlying medical conditions.

### 3.2. Hypothesis 2 (H2): Cognitive Scores Improve After First Stimulation Cycle

A significant difference between the baseline assessment and the post first stimulation cycle (6 sessions) can be seen used one-sided *t*-tests with alpha = 0.05: ADAS total: t (8) = 4.41 and *p* = 0.001 with *d* = 0.80, 95–KI (0.36, 1.25) and for ADAS Cog t (8) = 4.23, and *p* = 0.001 with *d* = 0.75, 95%-KI (0.33, 1.18) as shown in Figure 3A. No significant improvement could be seen in MMSE with t (9) = −0.97, *p* = 0.82 with *d* = −0.11, 95%-KI (−0.36, 0.13) and in MoCA t (9) = −4.30 *p* = 0.99 with *d* = −0.33, 95%-KI [−0.50, −0.16], however both scores show an incline in mean scores with MMSE *M*pre = 20.6 and *M*post = 21.4, and MoCA *M*pre = 14.6 and *M*post = 16.8.

Hypothesis 2 (H2) can be partly verified.

### 3.3. Hypothesis 3 (H3): Cognitive and Affective Scores Stay Stable After One Year of Treatment

Patients showed a stable score in all neuropsychological assessments during a one year period (MMSE: F(1) = 0.039, *p* = 0.843; R = 0.031; MoCA: F(1) = 0.031, *p* = 0.860; R = 0.030, ADAS Cog: F(1) = 0.167, *p* = 0.685; R = 0.073; ADAS: F(1) = 0.046, *p* = 0.832; R = 0.040) as shown in Figure 3A. For the MMSE, scores increased following the first stimulation cycle. At baseline, the mean score was 20.6 with a standard deviation (SD) of 6.50. After the first stimulation cycle, scores increased slightly to a mean of 21.4 (SD = 5.37). However, the scores returned to the baseline level at 3 months (M = 20.38, SD = 5.19) and remained stable at 6 months (M = 20.4, SD = 5.47) and 12 months (M = 20.1, SD = 6.89).

For the MoCA, there was an initial improvement following the first stimulation cycle. The mean score at baseline was 14.6 (SD = 6.10), which increased to 16.8 (SD = 5.49) after the first stimulation cycle and back to 15.16 (SD = 5.87) at 3 months. However, scores then declined to 12.75 (SD = 6.56) at 6 months before increasing back to the baseline score at M = 14.86 (SD = 6.34) at 12 months.

For the ADAS Cognitive, scores improved after the first stimulation cycle, with the mean score at baseline being 23.67 (SD = 6.62) and post-first stimulation cycle M = 18.56 (SD = 5.36), and further decreased to 15.0 (SD = 0.82) at 3 Months. However, there was a notable increase in scores at 6 Months (M = 20.0, SD = 0.00) and 12 Months (M = 25.8, SD = 12.33).

For the ADAS Total, scores decreased from baseline (M = 28.44, SD = 9.09) to 3 Months (M = 16.0, SD = 1.41). However, scores increased again at 6 months (M = 19.5, SD = 0.47) and further rose around baseline level at 12 months (M = 28.9, SD = 13.49).

Moreover, a trend of a decrease in depressive scores can be seen by assessing the mean scores of the patients for over a year (Figure 3B). Due to underpower, only mean scores are reported, and no statistical analysis was conducted. Before the first stimulation, the group of patients showed a mean score of 5.5 (SD = 6.97), and after a year, a mean score of 1.2 (0.47).

Hypothesis 3 (H3) can be verified. Additionally, improvement in depression was detected on a descriptive level.

### 3.4. Hypothesis 4 (H4): A Stronger Effect Can Be Seen in Memory and Speech Compared to Visuo-Construction Due to the Targeted Neuroanatomical Treatment Areas

We conducted longitudinal cognitive assessments to examine changes over time in distinct cognitive domains: memory, orientation, speech, and visuo-construction measured by the ADAS test. Lower scores indicate a better performance. The mean scores and standard deviations for each cognitive domain were computed at distinct time intervals: Baseline, Post 1. Stim Cycle, 3 Months, 6 Months, and 12 Months (see Figure 4).

Scores are reported in a descriptive quantitative manner.

For memory, a reduction in mean scores was observed over time. Initially, participants scored 12.7 (SD = 4.58) at baseline, which decreased to 10.55 (SD = 4.27) following the first stimulation cycle. At the 3-month mark, scores slightly increased to 11 (SD = 1). By the 12-month follow-up, mean scores rose to 11.9 (SD = 5.84).

For orientation, initially, participants scored 7.6 (SD = 6.63) at baseline, which decreased to 5.89 (SD = 4.46) post-stimulation cycle. At the 3-month evaluation, scores dropped to 2 (SD = 0). At the 12-month mark, the mean scores went up to 9 (SD = 6.99).

Regarding speech, an improvement, and, therefore, a deduction in mean scores was observed initially (baseline 4.2 (SD = 2.6)), which went to 1.67 (SD = 2.31) post-stimulation cycle. At 3 months, scores remained stable at 2 (SD = 1.5), then decreased to 0 (SD = 0) at 6 months. At the 12-month follow-up, mean scores were rising to 2.9 (SD = 3.53).

For visuo-construction, patients scored 0.9 (SD = 1.22) at baseline, which remained relatively stable post-stimulation cycle (1 (SD = 0.94)) and at 3 months (1 (SD = 0.47)), then changed to 0 (SD = 0.87) at 6 months. Nonetheless, at the 12-month assessment, mean scores were rising to 1.38 (SD = 1.86).

Summing up, hypothesis 4 (H4) can be verified at least on a descriptive level, although these improvements cannot be statistically verified.

### 3.5. Additional Post Hoc Analysis

To better understand the characteristics and potential reasons for patient dropout, we conducted a focused post hoc analysis on the largest subgroup of participants lost to follow-up, which consisted of those who discontinued the study shortly after the initial stimulation cycle (N = 10) (see Figure 1). Given that this was the most substantial group within our attrition data, we aimed to examine whether their early discontinuation was associated with a lack of treatment effect and/or specific adverse events.

Therefore, we performed a one-sided *t*-test on the MMSE scores of this group. The analysis yielded a statistically significant result: t (7) = −2.76, *p* = 0.014, indicating a significant improvement in this subset of patients.

To further investigate potential reasons for patient dropout, we analyzed the occurrence of adverse events within the loss to follow-up group after the first stimulation cycle (N = 10). Our analysis revealed that only one patient (10%) reported an adverse event (eye twitching after stimulation). However, this patient was unable to rate the severity of the event using the numerical rating scale (NRS). Considering that each patient received six stimulation sessions, the reported adverse event occurred in only one out of 60 sessions, corresponding to 1.67% of all treatment sessions.

## 4. Discussion

In this paper, we address safety, feasibility, and long-term effects of TPS in Alzheimer’s. We consider TPS treatment as safe with only transient AEs below 1% of the stimulation sessions.

However, AEs that at least have to be partly attributed to stimulation as ADE/ADR can not only occur focally as pain or unpleasant sensation but also systemically as behavioral changes or vigilance changes, especially in patients with underlying medical conditions. Vasodepressor (neurocardiogenic) syncope is a frequent response to anxiety and psycho-physical discomfort. It’s a commonly encountered phenomenon that was observed during TMS treatment [16]. Visceral discomfort, nausea, dizziness, and pallor are symptoms that were also observed in our data. Therefore, these events were considered at least partly device-related. Thus, we suggest that treatment centers should ensure that a patient couch is available in the treatment room so that the patient can be stabilized immediately in case of syncope. In addition, blood pressure test, lab tests, CT, ECG, and EEG should be available so that a differential diagnosis can be made quickly.

A(D)E/Rs mostly occurred within the period of the highest intensity of treatment within the first stimulation cycle, where six sessions were applied within two weeks. This goes along with the fact that the most improvement in cognition was seen within this two-week period.

Initially, 35 patients started TPS treatment. However, there was a large loss of follow-up. To better understand potential reasons for patient dropout, we conducted post hoc analyses on the largest loss to follow-up subgroup. Our findings show that this group still exhibited a significant effect in MMSE scores, suggesting that their discontinuation was not primarily due to a lack of treatment effect. Additionally, adverse events were rare, with only one patient (10%) reporting a mild, non-quantifiable adverse event (eye twitching), corresponding to just 1.67% of all treatment sessions. Given these results, we believe that other external factors, unrelated to treatment efficacy or safety, may have contributed to patient attrition. There are several reasons why some patients with Alzheimer’s disease do not continue the treatment with transcranial pulse stimulation. Logistical aspects, such as the need for regular visits to specialized centers, can be challenging for patients and caregivers, especially when transportation or mobility is limited. Recruitment of participants is another critical factor, as Alzheimer’s patients and their families may be hesitant to continue to participate due to concerns about the novelty of the intervention or uncertainty regarding its long-term efficacy. Additionally, acceptance of the intervention can vary; some patients may not perceive the expected benefits from the treatment, leading to frustration or loss of motivation to continue. Finally, the feasibility of TPS may be challenging for Alzheimer’s patients due to the method’s requirements. Session duration and frequency might be overwhelming for individuals with cognitive limitations, such as difficulty sitting still, tolerating the noise of the device, and following schedules. These challenges underline the importance of tailoring TPS protocols and addressing patient-specific needs to improve adherence and overall treatment outcomes.

A general improvement in neuropsychological test scores could be seen after the first stimulation cycle with six initial sessions, with a significant improvement tested in ADAS and ADAS cognitive score. No significance could be seen in the MMSE and MoCA scores, which might be caused by the lower sensitivity for mildly to moderately affected patients [17].

One could conclude that stimulation should be conducted with a higher frequency over a long-term period. However, due to practical reasons, only booster sessions once a month were done. This report confirms that Alzheimer’s patients treated regularly with that scheme show a stable score over a year of treatment.

Moreover, on a descriptive level, the data suggest that a slight improvement over time can be expected in memory, speech, and mood. This relates to brain areas that have been targeted with most stimulation pulses. Nevertheless, fluctuations over time—especially in ADAS scores as well as in memory and orientation domains—were observed and should be taken into account when interpreting the long-term treatment effects. These variations may reflect individual progression patterns, measurement variability, or nonspecific influences. Due to the limited sample size of ten patients with full one-year follow-up, we refrained from conducting subgroup analyses (e.g., mild vs. moderate AD). With such small subsamples, results would lack statistical power and risk being misleading. Future studies with larger cohorts are needed to explore whether patient characteristics such as disease stage influence the response to TPS treatment.

Potential limitations of this paper include the small sample size, lack of a control group, and variations in disease progression among participants. In addition, the high dropout rate may introduce a risk of attrition bias, as patients who continued treatment might differ systematically from those lost to follow-up. This limitation should be considered when interpreting the findings.

Nevertheless, in light of previous meta-analytic data showing a natural ADAS score worsening of approximately five points per year in Alzheimer’s disease [8], the observation of stable cognitive scores in our cohort suggests a potentially beneficial treatment effect of TPS over the long term, even in the absence of further cognitive improvement. These findings are further supported by Fong et al. (2023) [18], who demonstrated cognitive improvements following TPS in patients with mild neurocognitive disorder. Their results confirm that TPS may already exert beneficial effects at earlier disease stages, complementing our findings in patients with more advanced AD. Another limitation is that the individual variability in treatment response—potentially influenced by anatomical, genetic, or psychological factors—was not systematically assessed in this study and should be addressed in future research. In addition, raters were not blinded to the study aims, which may have introduced bias—particularly in domains involving subjective judgment such as speech evaluation.

One could argue that TPS effects on metabolism are more global and not target-specific. However, technically, TPS can be used in a highly focal manner by the use of neuronavigation. For the focal approach, it should be considered targeting the exact anatomical areas where patients show deficits. A study showed that sparing out stimulation areas that are essential for visuo-construction, these functions declined in a neuropsychological test [6], which is also presented by our findings. Thus, specific targeting makes sense, and one could concentrate on specific symptoms patients want to improve, including memory, speech, visuo-construction, and orientation.

Beyond clinical outcomes, the mechanisms by which TPS may delay the progression of Alzheimer’s disease deserve further exploration. Previous studies have suggested that TPS might promote neural plasticity and modulate functional network connectivity. For example, Wojtecki et al. (2025) [19] demonstrated EEG-based changes in oscillatory network activity following TPS, which may reflect increased functional connectivity and plastic reorganization. In addition, imaging studies have provided evidence for structural and metabolic changes. Popescu et al. (2021) [5] reported reduced cortical atrophy in patients who responded to TPS treatment. Recent evidence from a randomized, sham-controlled trial [4] further supports the functional effects of TPS in AD. Using functional MRI, the study demonstrated increased memory-related brain activation in regions such as the precuneus, frontal, and visual cortices after verum TPS. Additionally, resting-state functional connectivity within the dorsal attention network was significantly enhanced. These findings suggest that TPS may modulate brain network activity and cognitive functions via mechanisms of neuroplasticity. Interestingly, the cognitive benefits were most pronounced in younger patients (≤70 years), indicating a potential interaction between age and treatment responsiveness. These results align with our interpretation that TPS-induced improvements may be mediated by plastic reorganization in targeted brain networks.

For a general protocol in AD, the number of pulses in the targeted regions should be adjusted according to the protocol used to date. To see an effect in all cognitive domains that are affected by AD, we suggest that the protocol should be changed in the following way: In addition to the bilateral frontal lobe, bilateral parietal lobe, bilateral temporal lobe, and precuneus, the occipital lobe should also be targeted to enhance visoconstructive functions. Neuronavigation should be improved technically to visualize distinct deeper temporal areas of the hippocampus to improve memory and orientation. Consequently, this augmented protocol is proposed to be termed F-TOP^2^ (frontal–temporal–occipital–parietal–precuneus) and could have an overall effect.

Furthermore, as discussed in the introduction, the underlying effects of TPS are still not discovered. This paper shows a deduction of depressive symptoms, and three other studies could also show this effect [3,7]. Improvement in Alzheimer’s cognitive symptoms through the application of TPS may be attributed to its mood-enhancing effects, as heightened mood can lead to various benefits in managing the condition. Therefore, a global or as here proposed, a multifocal stimulation approach including wide areas of the frontal cortex seems plausible (Figure 5). Elevated mood can potentially enhance cognitive function and memory retention, reduce stress and anxiety levels, promote better sleep quality, increase engagement in social interactions and activities, and improve overall quality of life for individuals with Alzheimer’s disease. These factors collectively contribute to a more favorable environment for managing and alleviating Alzheimer’s symptoms. While our study focused primarily on cognitive outcomes, the improvements in mood and speech suggest that the effects of TPS may extend beyond strictly cognitive domains. The improvements observed in cognitive and mood domains may point to the potential of TPS to enhance socio-cognitive resilience in neurodegenerative populations. Recent studies have shown that non-invasive brain stimulation (NIBS) techniques, including rTMS, have the capacity not only to address symptom-specific deficits but also to support the development of broader adaptive capacities, such as emotional regulation and social cognition [20,21]. These findings underscore the importance of considering how NIBS might foster adaptive traits in addition to symptom remediation.

Therefore, the effects of TPS need to be further investigated so that the most beneficial treatment protocol for patients with Alzheimer’s can be developed.

## 5. Conclusions

This study provides the first long-term data on the feasibility, safety, and potential cognitive benefits of transcranial pulse stimulation (TPS) in Alzheimer’s disease. Our findings suggest that TPS is a generally well-tolerated treatment with a low rate of adverse events. Notably, improvements were observed in cognitive domains such as memory and speech after the initial stimulation cycle and remained stable over one year of monthly booster sessions. While individual fluctuations occurred, a general stabilization in cognitive decline—contrasting the expected disease progression—indicates a potential benefit of TPS as an add-on therapy. Furthermore, the observed mood improvements point toward broader effects beyond cognition, potentially supporting socio-emotional resilience. Given the anatomical specificity of TPS, we propose a novel multifocal stimulation approach (F-TOP2) targeting additional brain regions to enhance therapeutic outcomes across diverse cognitive domains. Future studies with larger cohorts and controlled designs are warranted to confirm these findings and optimize stimulation protocols.

## Figures and Tables

**Figure 1 brainsci-15-00830-f001:**
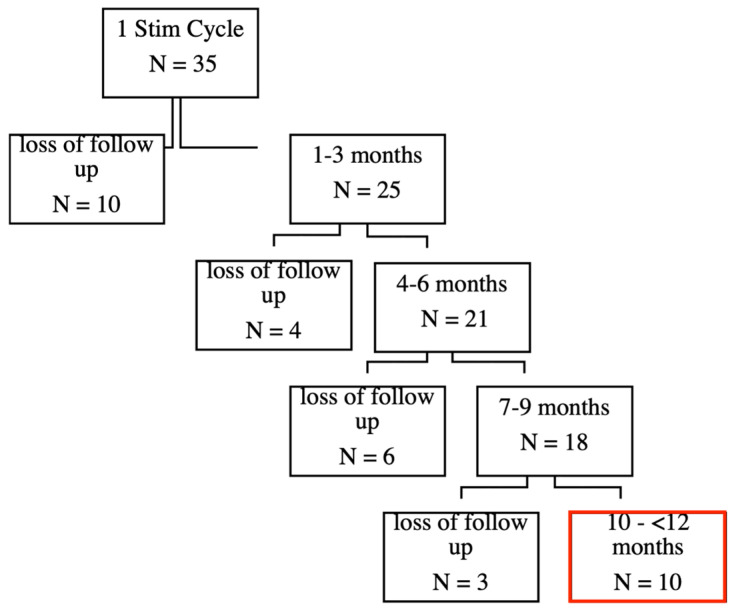
An overview of the patient numbers across the follow-up period. The majority of attrition occurred within the first months after the initial stimulation cycle.

**Figure 2 brainsci-15-00830-f002:**
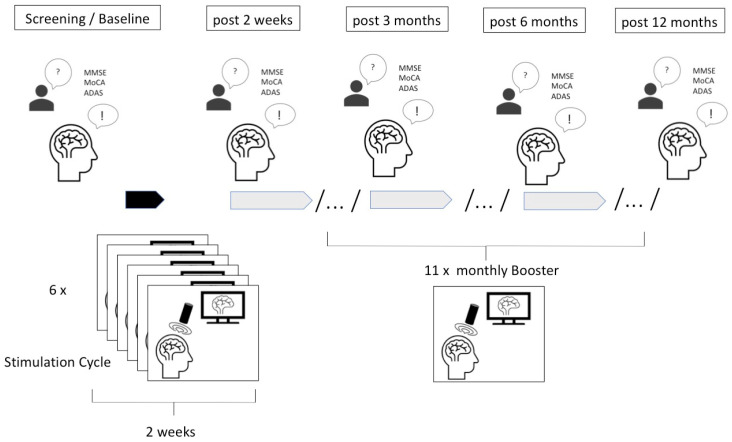
Timeline of the treatment and clinical assessments. First, the patient was screened for in-and exclusion criteria for treatment, followed by baseline assessment (including MMSE, MoCA, ADAS, ADAS Cog), the first six initial sessions of transcranial pulse stimulation within 2 weeks, post-stimulation assessment (including the same neuropsychological test battery as the baseline assessment), and monthly booster sessions with 6000 pulses each. Finally, there are assessments at 3, 6, and 12 months (including MMSE, MoCA, ADAS, and ADAS Cog; each ± 1 month). The stimulation treatments are shown in green and the neuropsychological assessments in blue. Note that BDI-II FS was also assessed in some patients but not for the whole group, and was not used for statistics.

**Figure 3 brainsci-15-00830-f003:**
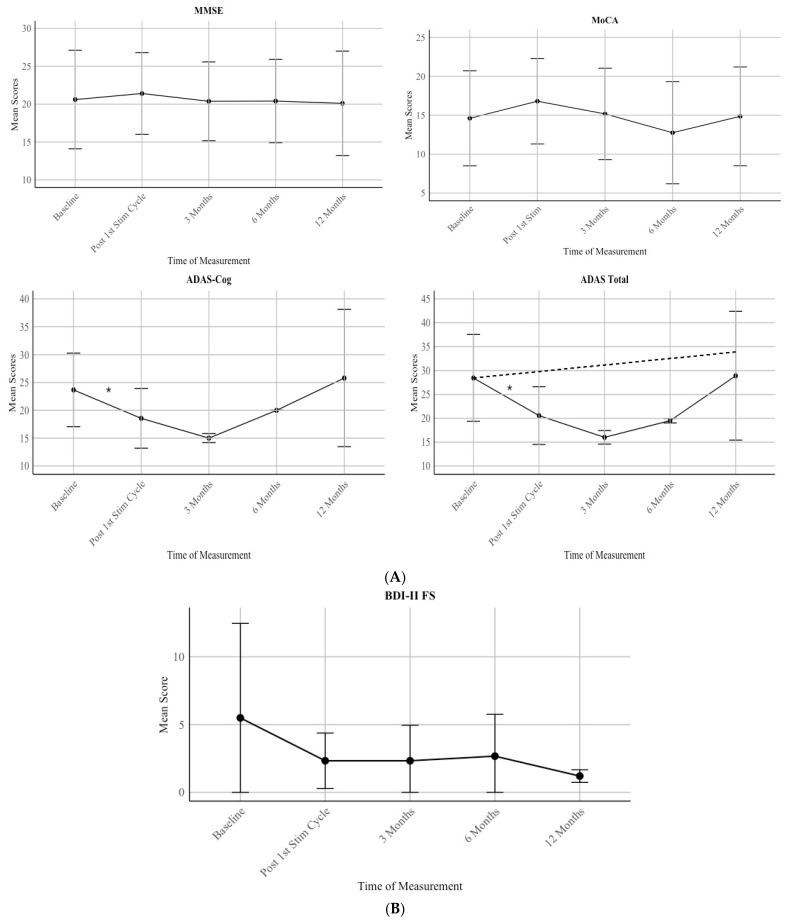
(**A**) Scores over time. MMSE. MoCA. ADAS_Cog, and ADAS_Total. The dashed line represents the natural disease progression found by a recent meta-analysis [8]. (**B**) BDI-II FS scores over time. Error bars mark SD. Asterisk (*) marks significant difference between the baseline assessment and the post first stimulation cycle used one-sided *t*-tests with *p* = 0.001 while scores stay table over time.

**Figure 4 brainsci-15-00830-f004:**
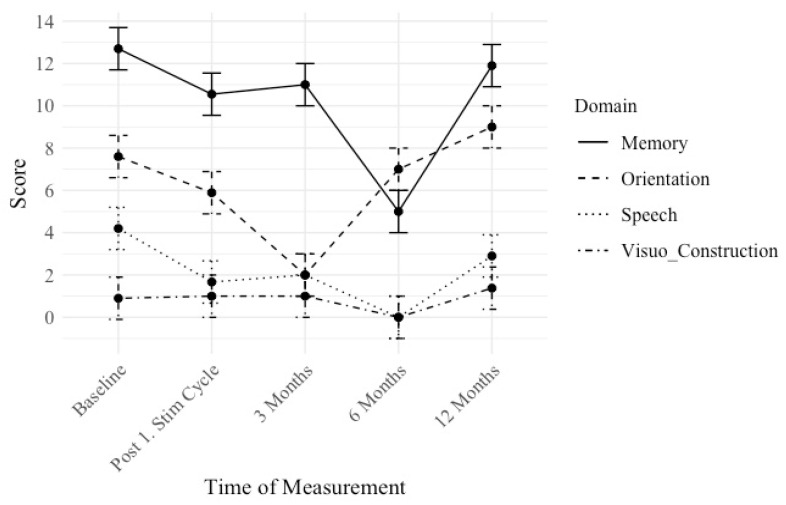
Mean scores for the different cognitive domains: memory, orientation, speech, and visuo-construction, assessed by the ADAS subscores over time. A lower score indicates a better performance. Memory and speech scores progress improve over time, while orientation and visuo-construction worsen slightly.

**Figure 5 brainsci-15-00830-f005:**
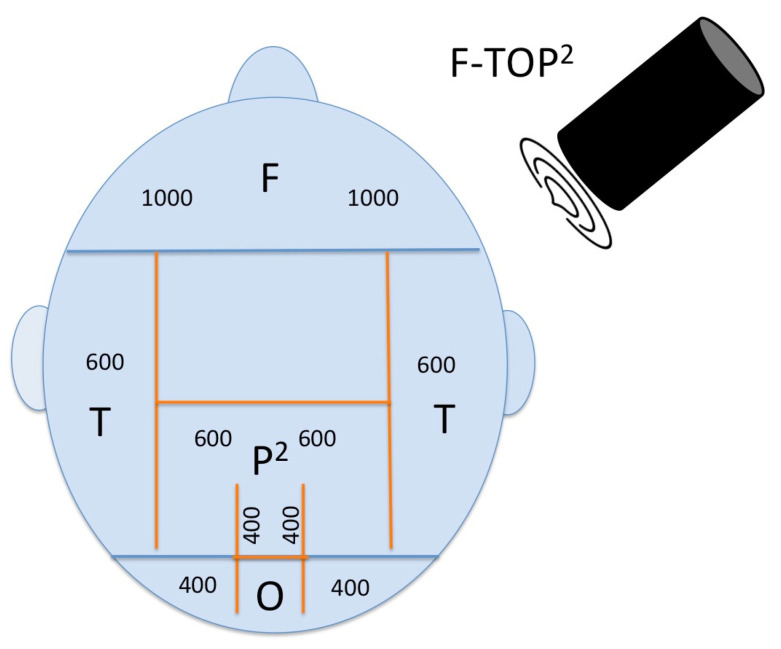
Proposed multifocal stimulation approach according to the F-TOP^2^ scheme. Number of pulses delivered by the navigated TPS application to the following areas: (F)rontal 1000 each hemisphere, (T)emporal 600 each hemisphere, (O)ccipital 400 each hemisphere, (P)arietal 600 each hemisphere, and (P)recuneus midline 800.

**Table 1 brainsci-15-00830-t001:** Demographics of the included patients.

ID	Age	Sex	Cognitive Impairment	Biomarkers Category/Diagnosis
1	76	m	Moderate	A+T+(N)+/AD
2	77	m	Moderate	Alzheimer’s Clinical Syndrome without biomarkers tested ^a^
3	59	m	Moderate	A+T-(N)+ ^b^
4	72	m	Mild	A+T+(N)+/AD
5	64	m	Moderate	A+T+(N)+/AD
6	76	m	Mild	A-T-(N)+ ^c^
7	63	f	Mild	A+T+(N)+/AD
8	79	m	Moderate	A+T+(N)+/AD
9	79	m	Mild	A+T+(N)+/AD
10	62	f	Mild	A+T-(N)+ ^b^

Cognitive impairment was defined using the Mini-Mental Status Examination (MMSE): 30–27, no impairment, 26–20, mild impairment, 19–10, moderate impairment, and <10, severe impairment. Diagnostic criteria were assessed according to the NIA-AA criteria. “A” labels biomarker of Aß plaques, “T” labels biomarkers of fibrillar tau, and “N” labels biomarkers of neurodegeneration or neuronal injury [15]. ^a^ One patient was included without biomarkers tested. ^b^ Alzheimer’s and concomitant suspected non-Alzheimer’s pathological change. ^c^ Alzheimer’s clinical syndrome with non-Alzheimer’s pathological change.

**Table 2 brainsci-15-00830-t002:** Reported A(D)E/Rs subjectively rated by the patient in brackets with the NRS (0–10, with 10 being highest severity) during the first stimulation cycle (6 sessions), the first 1–3 months, during 3–6 months of the stimulation, and during 6–12 months. * Two AEs rated as severe both by the patient and the investigator occurred in two patients and were partly causally related to TPS (ADE/ADR) but partly related to underlying medical conditions.

ID	First Stim. Cycle	1–3 Mo.	3–6 Mo.	6–12 Mo.
1	None	None	Nervousness [1,4]	None
2	Feeling of pressure/slight pain on the head during stimulation (2)	None	None	None
3	Pain in the jaw (4)	None	None	Syncope, vomiting during stimulation, combined with hypotension (10) *
4	Somnolence after stimulation with hypotension (8–10) *	None	None	Three days after stimulation, aggressive behavior and vigilance fluctuations (6)
5	None	None	None	None
6	None	None	None	None
7	None	None	None	None
8	None	None	None	None
9	None	Unpleasant focal sensation during stimulation (5)	None	None
10	Dizziness for two hours after stimulation (3)	None	None	None

## Data Availability

The datasets generated during and/or analyzed are available from the corresponding author on reasonable request. The datasets are not shared publicly to complie with the Ethics Committee and data safety approval.

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
