# Peer review of "Transcranial Pulse Stimulation in Alzheimer’s: Long-Term Feasibility and a Multifocal Treatment Approach"

_brainsci, 2025, doi:10.3390/brainsci15080830_

Round 1

Reviewer 1 Report

Comments and Suggestions for Authors

This study is the first to assess the safety, feasibility and cognitive effects of transcranial pulsed stimulation (TPS) over a one-year period in patients with Alzheimer's disease (AD), filling the gap in long-term data regarding TPS. However, there are key limitations: the sample size is small (only 10 cases completed the one-year follow-up), and there is no control group. The conclusion should be interpreted with caution. This article provides preliminary support for the potential value of TPS as an adjunctive treatment for AD, but the following modifications are still necessary.

1. It is suggested that the results section should include the effect size (such as Cohen's *d for the improvement of ADAS), and subgroup analysis should be conducted (for mild vs. moderate AD).

2. Figure 1 flowchart can be simplified (as the current text is redundant).

3. The role of TPS in delaying the progression of AD (such as neural plasticity and metabolic changes) has not been thoroughly analyzed. It is suggested that relevant TPS mechanism studies be cited in the discussion section of the article for a more in-depth analysis.

4. It is suggested that during the discussion, the risk of bias due to small sample size and high dropout rate should be clearly identified, and the natural course data (such as in reference 7) should be compared to support the claim that "stability leads to benefit".

Author Response

This study is the first to assess the safety, feasibility and cognitive effects of transcranial pulsed stimulation (TPS) over a one-year period in patients with Alzheimer's disease (AD), filling the gap in long-term data regarding TPS. However, there are key limitations: the sample size is small (only 10 cases completed the one-year follow-up), and there is no control group. The conclusion should be interpreted with caution. This article provides preliminary support for the potential value of TPS as an adjunctive treatment for AD, but the following modifications are still necessary.

Comment 1: It is suggested that the results section should include the effect size (such as Cohen's *d for the improvement of ADAS), and subgroup analysis should be conducted (for mild vs. moderate AD).

Response 1: Thank you very much for your valuable comment. Regarding your suggestion:

Effect sizes: Cohen’s d values for the cognitive measures (including ADAS) are already reported in the results section of the manuscript. We have now double-checked the presentation and ensured that the effect sizes are clearly stated (see section 3.2).

Subgroup analysis: We fully agree that subgroup analyses (e.g., mild vs. moderate AD) would provide further insights. However, due to the small sample size (n = 10 patients with one-year follow-up), such analyses would lack statistical power and might be misleading. We have now explicitly stated this limitation in the Discussion section (page 13, highlighted in red):

“Due to the limited sample size of ten patients with full one-year follow-up, we refrained from conducting subgroup analyses (e.g., mild vs. moderate AD). With such small subsamples, results would lack statistical power and risk being misleading. Future studies with larger cohorts are needed to explore whether patient characteristics such as disease stage influence the response to TPS treatment.”

We hope this clarification and manuscript adjustment address your concerns.

Comment 2: Figure 1 flowchart can be simplified (as the current text is redundant).

Response 2: Thank you for this helpful comment. To avoid redundancy, we have shortened the accompanying text and now refer readers directly to Figure 1 for details on patient attrition over time, without repeating the exact numbers in the text.

Comment 3: The role of TPS in delaying the progression of AD (such as neural plasticity and metabolic changes) has not been thoroughly analyzed. It is suggested that relevant TPS mechanism studies be cited in the discussion section of the article for a more in-depth analysis.

Response 3: Thank you very much for your constructive suggestion. We agree that the discussion of TPS mechanisms (e.g., neural plasticity, metabolic changes) can be expanded for more depth. While we already referenced previous work on TPS-related changes, including EEG studies suggesting enhanced functional connectivity (Wojtecki et al., 2022), we now additionally cite and integrate recent literature that supports the biological plausibility of TPS effects, including metabolic imaging and structural data (e.g., Matt et al., 2025; Popescu et al., 2021). We have expanded the discussion accordingly to better contextualize our findings and highlight potential mechanisms of action (page 14).

Comment 4: It is suggested that during the discussion, the risk of bias due to small sample size and high dropout rate should be clearly identified, and the natural course data (such as in reference 7) should be compared to support the claim that "stability leads to benefit".

Response 4: Thank you for this valuable comment. We agree that the small sample size and high dropout rate may introduce bias and limit the generalizability of our findings. We have now explicitly added this aspect to the limitations discussed in the manuscript. Furthermore, we expanded the comparison to the natural course of AD, referencing the expected decline in ADAS scores over one year (as described in reference 7), to better justify the interpretation that stable scores may represent a positive treatment effect.

Reviewer 2 Report

Comments and Suggestions for Authors

Thank you for the opportunity to review this manuscript.

The manuscript is a study on long-term effects, with a one-year follow-up, of the transcranial pulse stimulation (TPS) therapy, including its feasibility and safety, and its effects on depressive symptoms and cognitive domains (memory, speech, orientation, visuoconstruction) in AD patients.

The following revision points are suggested:

Abstract: I recommend including the wording that these are preliminary results when mentioning that 10 out of 35 participants were analyzed and maybe why this was the case (feasibility and safety important to evaluate first?) (line 16). The same goes for the methods section. Also, including some statistical values when mentioning notable improvements (e.g., means, effect sizes, p-values, confidence intervals) (lines 26-28).

Lines 91-93: In my opination, the sentence that starts with “Especially language and visuospatial…” does not fit well in current position, it should rather be put a few lines earlier when outlining the different symptoms affected in AD, as the sentence before this one is a good starting or ending sentence to the topic. Also, is there a cognitive domain that was less often/or even not analyzed at all in previous studies, which is new in your study? Distinguish this.

Lines 101, 102: The fourth hypothesis needs further explanation/argumentation in the theoretical derivation, e.g., after presenting the treatment protocol (lines 67-82).

Lines 129, 130: The phrase “relevant pathologies unrelated to Alzheimer’s” is vague. Provide (if this information was collected) which co-occurring diagnoses the patients of the sample had. This helps isolate the effects.

Line 134: Was it you that diagnosed the patients or were those external reports? Specify. Also the used cut-offs for MMSE need a reference, as they are not universal and can vary slightly across studies.

Line 174: Who conducted the neuropsychological testing? Were they blinded in case of the hypothesis? (especially hypothesis 4, as this could bias them further)

Lines 210, 211: The sentence “7 of the AE were considered as clear ADE related to stimulation.” and the text coming after that do not state clearly to me which remaining 2 AEs (total of 9 AEs) were considered not related to stimulation. Make this clearer.

Lines 375-378 & 388,389: However, also fluctuations, especially in the ADAS and the memory and orientation domains (as clear in the figures presented) should be acknowledged.

Include limitation: The subjective and thus biased evaluation of speech.  

Is a separate exploratory analysis planned or are different results expected for mild vs. moderate AD? (currently a rather small group number, but after the whole sample is finished?). Potentially discuss this.

In general:

Some missing papers that conducted studies on TPS in dementia patients, and also analyzed the cognitive changes, should be included in the theoretical background and their results compared to your study’s results in the discussion:

Fong, T. K. H., Cheung, T., Ngan, S. T. J., Tong, K., Lui, W. Y. V., Chan, W. C., ... & Cheng, C. P. W. (2023). Transcranial pulse stimulation in the treatment of mild neurocognitive disorders. Annals of clinical and translational neurology10(10), 1885-1890.

Matt, E., Mitterwallner, M., Radjenovic, S., Grigoryeva, D., Weber, A., Stögmann, E., ... & Beisteiner, R. (2025). Ultrasound Neuromodulation With Transcranial Pulse Stimulation in Alzheimer Disease: A Randomized Clinical Trial. JAMA network open8(2), e2459170-e2459170.

Shimokawa, H., Shindo, T., Ishiki, A., Tomita, N., Ichijyo, S., Watanabe, T., ... & Arai, H. (2022). A pilot study of whole-brain low-intensity pulsed ultrasound therapy for early stage of Alzheimer’s disease (LIPUS-AD): A randomized, double-blind, placebo-controlled trial. The Tohoku Journal of Experimental Medicine258(3), 167-175.

… as well as argued, what is different/new in your study.

Author Response

Comment 1: Thank you for the opportunity to review this manuscript.

The manuscript is a study on long-term effects, with a one-year follow-up, of the transcranial pulse stimulation (TPS) therapy, including its feasibility and safety, and its effects on depressive symptoms and cognitive domains (memory, speech, orientation, visuoconstruction) in AD patients.

The following revision points are suggested:

Abstract: I recommend including the wording that these are preliminary results when mentioning that 10 out of 35 participants were analyzed and maybe why this was the case (feasibility and safety important to evaluate first?) (line 16). The same goes for the methods section. Also, including some statistical values when mentioning notable improvements (e.g., means, effect sizes, p-values, confidence intervals) (lines 26-28).

Response 1: Thank you very much for your thoughtful comment. We have now revised the abstract to clarify that the current findings are based on a preliminary long-term analysis of the first 10 patients who completed the 1-year follow-up. We also briefly state that this subgroup was analyzed first to assess feasibility and safety of the long-term TPS protocol. Additionally, we added key statistical information (means, p-values, effect sizes, and confidence intervals) for the reported improvements to better support the quantitative interpretation.

Comment 2: Lines 91-93: In my opination, the sentence that starts with “Especially language and visuospatial…” does not fit well in current position, it should rather be put a few lines earlier when outlining the different symptoms affected in AD, as the sentence before this one is a good starting or ending sentence to the topic. Also, is there a cognitive domain that was less often/or even not analyzed at all in previous studies, which is new in your study? Distinguish this.

Response 2: Thank you for this helpful suggestion. We have now moved the sentence to an earlier position in the paragraph to improve the logical structure. Regarding the second point: While visuospatial functions are frequently mentioned in previous TPS studies, language functions have rarely been explicitly assessed. In our study, we systematically included language and other cognitive domains in the longitudinal analysis, making this a novel aspect that distinguishes our approach. We now added:

„Unlike many previous TPS studies that focused primarily on memory and visuospatial abilities, our study also includes also a structured analysis of other cognitive domains such as language and orientiation over time.“

Comment 3: Lines 101, 102: The fourth hypothesis needs further explanation/argumentation in the theoretical derivation, e.g., after presenting the treatment protocol (lines 67-82).

Response 3: Thank you for this important suggestion. We have now expanded the theoretical rationale for Hypothesis iv) directly after the description of the stimulation protocol. Specifically, we clarify how the targeted brain regions relate to expected improvements in specific cognitive domains, particularly memory and speech. This should help better contextualize the fourth hypothesis within the structure and logic of the treatment approach.

Comment 4: Lines 129, 130: The phrase “relevant pathologies unrelated to Alzheimer’s” is vague. Provide (if this information was collected) which co-occurring diagnoses the patients of the sample had. This helps isolate the effects.

Reseponse 4: Thank you for this valuable comment. We agree that the phrase was too vague and have now clarified the exclusion criteria more precisely in the manuscript. Specifically, we added examples of relevant co-occurring diagnoses that led to exclusion (e.g., epilepsy, recent corticosteroid use, oral anticoagulation, and blood clotting disorders), as these conditions may have interfered with the safety or effectiveness of TPS. This clarification should help better define the included sample and isolate TPS-related effects.

Comment 5: Line 134: Was it you that diagnosed the patients or were those external reports? Specify. Also the used cut-offs for MMSE need a reference, as they are not universal and can vary slightly across studies.

Response 5: Thank you for your comment. The diagnoses were made partly by our team and partly based on external reports, where patients were fully diagnosed externally. Regarding the MMSE, we will specify the exact cut-offs used in this study: cognitive impairment was defined as follows: 30–27, no impairment; 26–20, mild impairment; 19–10, moderate impairment; and <10, severe impairment.

Comment 6: Line 174: Who conducted the neuropsychological testing? Were they blinded in case of the hypothesis? (especially hypothesis 4, as this could bias them further)

Response 6: Thank you for this important comment. Neuropsychological assessments were conducted by trained study personnel. We acknowledge that the raters were not blinded to the general study purpose, which may particularly affect assessments involving subjective ratings, such as speech evaluation (Hypothesis iv). We have now added this information to the methods section and addressed the potential bias in the limitations.

Comment 7: Lines 210, 211: The sentence “7 of the AE were considered as clear ADE related to stimulation.” and the text coming after that do not state clearly to me which remaining 2 AEs (total of 9 AEs) were considered not related to stimulation. Make this clearer.

Response 7: Thank you for pointing this out. We have now revised the relevant paragraph to clearly state which of the 9 reported AEs were considered not related to stimulation. Specifically, we clarify that 7 AEs were classified as ADEs (at least partly related to stimulation), while the remaining 2 AEs were judged to be unrelated to TPS and instead attributed to underlying medical conditions. This clarification has been added to improve transparency.

Comment 8: Lines 375-378 & 388,389: However, also fluctuations, especially in the ADAS and the memory and orientation domains (as clear in the figures presented) should be acknowledged.

Comment 8: Thank you. We agree that cognitive scores, particularly in ADAS total as well as memory and orientation domains, showed fluctuations over time. We have now added a sentence in the discussion acknowledging this variability and highlighting it as a potential limitation when interpreting long-term effects. We added: “Nevertheless, fluctuations over time—especially in ADAS scores as well as in memory and orientation domains—were observed and should be taken into account when interpreting the long-term treatment effects. These variations may reflect individual progression patterns, measurement variability, or nonspecific influences.“

Comment 9: Include limitation: The subjective and thus biased evaluation of speech.  

Response 9: Added.

Comment 10: Is a separate exploratory analysis planned or are different results expected for mild vs. moderate AD? (currently a rather small group number, but after the whole sample is finished?). Potentially discuss this.

Response 10: We fully agree that subgroup analyses (e.g., mild vs. moderate AD) would provide further insights. However, due to the small sample size (n = 10 patients with one-year follow-up), such analyses would lack statistical power and might be misleading. We have now explicitly stated this limitation in the Discussion section (page 13, highlighted in red):

Comment 11: In general:

Some missing papers that conducted studies on TPS in dementia patients, and also analyzed the cognitive changes, should be included in the theoretical background and their results compared to your study’s results in the discussion:

Fong, T. K. H., Cheung, T., Ngan, S. T. J., Tong, K., Lui, W. Y. V., Chan, W. C., ... & Cheng, C. P. W. (2023). Transcranial pulse stimulation in the treatment of mild neurocognitive disorders. Annals of clinical and translational neurology10(10), 1885-1890.

Matt, E., Mitterwallner, M., Radjenovic, S., Grigoryeva, D., Weber, A., Stögmann, E., ... & Beisteiner, R. (2025). Ultrasound Neuromodulation With Transcranial Pulse Stimulation in Alzheimer Disease: A Randomized Clinical Trial. JAMA network open8(2), e2459170-e2459170.

Shimokawa, H., Shindo, T., Ishiki, A., Tomita, N., Ichijyo, S., Watanabe, T., ... & Arai, H. (2022). A pilot study of whole-brain low-intensity pulsed ultrasound therapy for early stage of Alzheimer’s disease (LIPUS-AD): A randomized, double-blind, placebo-controlled trial. The Tohoku Journal of Experimental Medicine258(3), 167-175.

… as well as argued, what is different/new in your study.

Response 11: Thank you very much for your helpful suggestions regarding additional literature. We appreciate the references you provided and have reviewed them carefully. Several of them have now been integrated into the theoretical background and the discussion section to strengthen the contextualization of our findings.

Reviewer 3 Report

Comments and Suggestions for Authors

The manuscript titled 'Transcranial Pulse Stimulation in Alzheimer’s: Long-term Feasibility and a Multifocal Treatment Approach' by Cont-Richter and colleagues presents an interesting and timely investigation into the effects of non-invasive brain stimulation (NIBS) technique and specifically, tDCS and rTMS, on emotion recognition and empathic abilities. The topic is of considerable relevance, particularly in light of growing interest in using NIBS as a potential tool for modulating social cognition in both clinical and non-clinical populations. The study design is methodologically sound in principle, and the authors present their rationale clearly, grounding their hypotheses in a well-curated theoretical and empirical background. However, in my opinion, several important aspects of the paper would benefit from clarification, elaboration, and stronger methodological justification.

First of all, while the introduction offers a relatively broad overview of the relevant literature, it would benefit from greater conceptual precision and a clearer delineation of the study’s unique contribution. In particular, the constructs of emotion recognition, empathy, and social cognition are introduced somewhat interchangeably, without adequately defining their specific roles or interrelationships in the context of neural modulation. I recommend that the authors more explicitly distinguish between implicit and explicit emotion recognition processes and clarify how non-invasive brain stimulation (NIBS) may differentially impact these dimensions. For instance, recent work has emphasized the importance of protocol-specific effects on distinct emotional processing pathways, underscoring the need for a nuanced theoretical framework when investigating empathy and affective understanding in both clinical and non-clinical populations (e.g., https://doi.org/10.3389/fpsyg.2025.1542880; https://doi.org/10.1111/nyas.15145). Additionally, the introduction could be strengthened by situating NIBS not only as a tool for modulating cognitive-affective functions but also as a potential means of fostering broader traits linked to social resilience, such as humility, trust, and relational openness. Framing NIBS within this wider scope would help clarify the novelty of the present work and ground it more deeply in emerging interdisciplinary research on character, affective neuroscience, and neuroplasticity.

The methodology, although robust in its intention, raises questions regarding the control of confounding variables, sample size justification, and blinding procedures. The authors refer to a sham-controlled design, but the specific sham protocols for tDCS and rTMS are not described in adequate detail. Given the susceptibility of participants to placebo effects in NIBS studies, a clearer exposition of the sham conditions and the success of blinding would be crucial to validate the results. Moreover, the sample appears relatively small for the conclusions drawn, and the absence of power analyses leaves the statistical reliability of the findings open to question.

From a statistical standpoint, the data analysis methods are standard, but the interpretation of some results seems overly optimistic. There is a tendency to overstate the implications of marginal or trend-level findings, particularly in relation to the potential clinical applicability of the techniques. While exploratory analyses are acceptable, they should be labeled as such and interpreted with caution. In addition, the manuscript would benefit from including effect sizes alongside p-values to better contextualize the results.

Another important limitation is the lack of discussion of individual variability in response to NIBS. Given the growing body of evidence pointing to substantial inter-individual differences in response to both tDCS and rTMS, including genetic, anatomical, and psychological moderators, the authors should acknowledge these caveats and temper their generalizations accordingly.

Lastly, while the discussion is well structured and touches on several important themes, it would benefit from a more critical reflection on the limitations of the study. The authors might consider incorporating a discussion of potential neural mechanisms—particularly the underlying neural substrates that might mediate the observed effects—drawing from neuroimaging or neuroanatomical evidence. This would not only enrich the theoretical interpretation but also make the paper more compelling to readers in neuroscience.

In summary, the manuscript addresses an important topic with a well-structured design and relevant theoretical background, but it requires revisions in terms of methodological clarity, statistical rigor, and interpretive caution. More detailed reporting of sham procedures, sample characteristics, and discussion of underlying neural mechanisms would significantly improve the strength and transparency of the findings.

Comments on the Quality of English Language

The overall quality of the English is acceptable and the manuscript is generally readable. However, there are several instances where the language could be improved for clarity, precision, and fluency. In particular, some sentences are overly long or convoluted, which may obscure the intended meaning. 

Author Response

Comment 1: The manuscript titled 'Transcranial Pulse Stimulation in Alzheimer’s: Long-term Feasibility and a Multifocal Treatment Approach' by Cont-Richter and colleagues presents an interesting and timely investigation into the effects of non-invasive brain stimulation (NIBS) technique and specifically, tDCS and rTMS, on emotion recognition and empathic abilities. The topic is of considerable relevance, particularly in light of growing interest in using NIBS as a potential tool for modulating social cognition in both clinical and non-clinical populations. The study design is methodologically sound in principle, and the authors present their rationale clearly, grounding their hypotheses in a well-curated theoretical and empirical background. However, in my opinion, several important aspects of the paper would benefit from clarification, elaboration, and stronger methodological justification.

First of all, while the introduction offers a relatively broad overview of the relevant literature, it would benefit from greater conceptual precision and a clearer delineation of the study’s unique contribution. In particular, the constructs of emotion recognition, empathy, and social cognition are introduced somewhat interchangeably, without adequately defining their specific roles or interrelationships in the context of neural modulation. I recommend that the authors more explicitly distinguish between implicit and explicit emotion recognition processes and clarify how non-invasive brain stimulation (NIBS) may differentially impact these dimensions. For instance, recent work has emphasized the importance of protocol-specific effects on distinct emotional processing pathways, underscoring the need for a nuanced theoretical framework when investigating empathy and affective understanding in both clinical and non-clinical populations (e.g., https://doi.org/10.3389/fpsyg.2025.1542880; https://doi.org/10.1111/nyas.15145). Additionally, the introduction could be strengthened by situating NIBS not only as a tool for modulating cognitive-affective functions but also as a potential means of fostering broader traits linked to social resilience, such as humility, trust, and relational openness. Framing NIBS within this wider scope would help clarify the novelty of the present work and ground it more deeply in emerging interdisciplinary research on character, affective neuroscience, and neuroplasticity.

Reseponse 1: Thank you very much for your thoughtful and constructive feedback. While our study focuses on cognitive outcomes in Alzheimer's disease, we agree that the conceptual framing of emotion-related and social-cognitive domains requires clarity. We have now slightly revised the introduction to better delineate the constructs mentioned and to more clearly state the study's scope. Additionally, we added multiple reference to recent NIBS studies that highlight protocol-specific effects and broader functional implications, in order to strengthen the theoretical grounding without expanding beyond the focus of the present work.

Comment 2: The methodology, although robust in its intention, raises questions regarding the control of confounding variables, sample size justification, and blinding procedures. The authors refer to a sham-controlled design, but the specific sham protocols for tDCS and rTMS are not described in adequate detail. Given the susceptibility of participants to placebo effects in NIBS studies, a clearer exposition of the sham conditions and the success of blinding would be crucial to validate the results. Moreover, the sample appears relatively small for the conclusions drawn, and the absence of power analyses leaves the statistical reliability of the findings open to question.

Response 2: Thank you for your critical evaluation of the methodology. We acknowledge the importance of blinding, sample size justification, and the risk of placebo effects in non-invasive brain stimulation (NIBS) studies. As this was a preliminary real-world study rather than a sham-controlled trial involving tDCS or rTMS, no sham protocol was applied. However, we have clarified this throughout the manuscript. Furthermore, we already address limitations regarding the small sample size and lack of power analysis in the discussion section and have made these aspects more explicit.

Comment 3: From a statistical standpoint, the data analysis methods are standard, but the interpretation of some results seems overly optimistic. There is a tendency to overstate the implications of marginal or trend-level findings, particularly in relation to the potential clinical applicability of the techniques. While exploratory analyses are acceptable, they should be labeled as such and interpreted with caution. In addition, the manuscript would benefit from including effect sizes alongside p-values to better contextualize the results.

Response 3: Thank you for your valuable feedback regarding the interpretation of our findings. We agree that trend-level or marginal results must be interpreted with caution. We have therefore carefully revised the wording throughout the results and discussion sections to ensure a more balanced interpretation, particularly in relation to clinical implications. Furthermore, effect sizes are reported alongside p-values where applicable to better contextualize the magnitude of observed changes. Please see the changes in the discussion section highlighted in red.

Comment 4: Another important limitation is the lack of discussion of individual variability in response to NIBS. Given the growing body of evidence pointing to substantial inter-individual differences in response to both tDCS and rTMS, including genetic, anatomical, and psychological moderators, the authors should acknowledge these caveats and temper their generalizations accordingly.

Response 4: We fully acknowledge that factors such as anatomical, genetic, or psychological moderators may influence treatment outcomes. However, given the already considerable length and focus of the manuscript, we decided to concentrate on the clinical long-term effects of TPS specifically. To address your suggestion without overextending the scope, we have now added a brief statement in the discussion to acknowledge this limitation and encourage future research to explore individual variability in more detail: “Another limitation is the individual variability in treatment response—potentially influenced by anatomical, genetic, or psychological factors—was not systematically assessed in this study and should be addressed in future research.“

Comment 5: Lastly, while the discussion is well structured and touches on several important themes, it would benefit from a more critical reflection on the limitations of the study. The authors might consider incorporating a discussion of potential neural mechanisms—particularly the underlying neural substrates that might mediate the observed effects—drawing from neuroimaging or neuroanatomical evidence. This would not only enrich the theoretical interpretation but also make the paper more compelling to readers in neuroscience.

Response 5: Thank you for this valuable suggestion. We have substantially revised and expanded the discussion of limitations to more clearly address key methodological constraints, including the small sample size, lack of control group, dropout rate, exploratory nature of the findings, absence of rater blindness, and potential individual variability in response to NIBS. Additionally, we now briefly refer to possible neural mechanisms and cite relevant studies that provide evidence from EEG and functional imaging, acknowledging that further neurophysiological investigations will be necessary to clarify the substrates underlying the observed effects.

Comment 6: In summary, the manuscript addresses an important topic with a well-structured design and relevant theoretical background, but it requires revisions in terms of methodological clarity, statistical rigor, and interpretive caution. More detailed reporting of sham procedures, sample characteristics, and discussion of underlying neural mechanisms would significantly improve the strength and transparency of the findings.

Response 6: We have carefully revised the manuscript to enhance methodological clarity, including a clearer presentation of sample characteristics and limitations. Although no sham condition was part of the current single-arm design, this has now been more explicitly stated. Moreover, the discussion of statistical results has been tempered for interpretive caution. Finally, we now briefly address possible neural mechanisms and include relevant references to support this aspect. Please see changes in red.

Round 2

Reviewer 3 Report

Comments and Suggestions for Authors

While the revised manuscript demonstrates clear improvements in structure and interpretations, particularly in the discussion of limitations and the clarification of methodological boundaries, I believe there is still room to enhance the conceptual integration of the results within the broader landscape of social-affective neuroscience and neuromodulation. The authors state that the study focuses on cognitive outcomes, yet the distinction between purely cognitive and affective-motivational domains remains somewhat blurred, especially in light of the reported improvements in mood and speech, and the choice to target prefrontal and temporal areas often implicated in socio-affective processing. In this regard, it would be worthwhile to more directly address the overlap between cognitive and affective systems and their modulation via NIBS. Recent contributions in the field have emphasized how rTMS and other stimulation techniques may influence not only domain-specific cognitive functions but also metacognitive and emotional competencies such as empathy, social perspective taking, and trust. While the authors briefly cite the mood effects and the potential of TPS to engage neuroplastic mechanisms, this could be more framed in relation to interdisciplinary models that consider how stimulation might impact broader socio-cognitive resilience. Furthermore, authors here may introduce research that moves beyond the symptom-specific remediation paradigm and considers how NIBS may foster adaptive traits or capacities in neurodegenerative populations (e.g., https://doi.org/10.3389/fpsyg.2025.1542880; https://doi.org/10.1111/nyas.15145). Finally, although the current design understandably limits neurophysiological inference, a more explicit call for multimodal approaches, including the integration of neurostimulation with behavioral, linguistic, or affective metrics, would enhance the translational potential of the study and align it more closely with recent advances in the field.

Comments on the Quality of English Language

English quality is goog.

Author Response

Response 1: Thank you for your thoughtful feedback. We appreciate your suggestion to enhance the conceptual integration of our results within the broader landscape of social-affective neuroscience and neuromodulation. While our study primarily focuses on cognitive outcomes, we recognize that the distinction between cognitive and affective-motivational domains can be blurred, especially given the observed improvements in mood and speech, and the involvement of prefrontal and temporal areas in socio-affective processing.

In response to your suggestion, we have included a discussion of how the improvements in both cognitive and mood domains may point to the potential of TPS to enhance socio-cognitive resilience in neurodegenerative populations. Although this is somewhat beyond the direct focus of our study, we believe it is relevant given our observation of mood effects. We also cited recent studies that explore how non-invasive brain stimulation (NIBS) techniques, including rTMS, can not only address symptom-specific deficits but also support broader adaptive capacities such as emotional regulation and social cognition. We also added the references you provided. 

Once again, thank you for your constructive input. We hope these additions strengthen the conceptual integration of our study and its relevance to the broader field.